# LOCAL LABEL PROPAGATION FOR LARGE-SCALE SEMI-SUPERVISED LEARNING

## ABSTRACT

A significant issue in training deep neural networks to solve supervised learning tasks is the need for large numbers of labeled datapoints. The goal of semi-supervised learning is to leverage ubiquitous unlabeled data, together with small quantities of labeled data, to achieve high task performance. Though substantial recent progress has been made in developing semi-supervised algorithms that are effective for comparatively small datasets, many of these techniques do not scale readily to the large (unlabeled) datasets characteristic of real-world applications. In this paper we introduce a novel approach to scalable semi-supervised learning, called Local Label Propagation (LLP). Extending ideas from recent work on unsupervised embedding learning, LLP first embeds datapoints, labeled and otherwise, in a common latent space using a deep neural network. It then propagates pseudolabels from known to unknown datapoints in a manner that depends on the local geometry of the embedding, taking into account both inter-point distance and local data density as a weighting on propagation likelihood. The parameters of the deep embedding are then trained to simultaneously maximize pseudolabel categorization performance as well as a metric of the clustering of datapoints within each psuedo-label group, iteratively alternating stages of network training and label propagation. We illustrate the utility of the LLP method on the ImageNet dataset, achieving results that outperform previous state-of-the-art scalable semi-supervised learning algorithms by large margins, consistently across a wide variety of training regimes. We also show that the feature representation learned with LLP transfers well to scene recognition in the Places 205 dataset.

## 1 INTRODUCTION

Deep neural networks (DNNs) have achieved impressive performance on tasks across a variety of domains, including vision (Krizhevsky et al., 2012; Simonyan & Zisserman, 2014; He et al., 2016a; 2017), speech recognition (Hinton et al., 2012; Hannun et al., 2014; Deng et al., 2013; Noda et al., 2015), and natural language processing (Young et al., 2018; Hirschberg & Manning, 2015; Conneau et al., 2016; Kumar et al., 2016). However, these achievements often heavily rely on large-scale labeled datasets, requiring burdensome and expensive annotation efforts. This problem is especially acute in specialized domains such as medical image processing, where annotation may involve performing an invasive process on patients.

Semi-supervised learning (SSL) seeks to learn useful representations from limited amounts of labeled data, leveraging it in conjunction with extensive unlabeled data. SSL has shown significant promise (Liu et al., 2018; Iscen et al., 2019; Zhai et al., 2019; Miyato et al., 2018; Tarvainen & Valpola, 2017; Lee, 2013; Grandvalet & Bengio, 2005; Qiao et al., 2018; Xie et al., 2019). However, gaps to supervised performance levels still remain significant, and many recent SSL methods rely on techniques whose efficiency scales poorly with dataset size and thus cannot be readily applied to many real-world machine learning problems (Liu et al., 2018; Iscen et al., 2019).

Here, we propose a novel SSL algorithm that is specifically adapted for use with large sparsely-labeled datasets. This algorithm, termed Local Label Propagation (LLP), learns a nonlinear embedding of the input data, and exploits the local geometric structure of the latent embedding space to help infer useful pseudo-labels for unlabeled datapoints. LLP borrows the framework of non-parametric embedding learning, which has recently shown utility in unsupervised learning (Wu et al., 2018b;

Zhuang et al., 2019), to first train a deep neural network that embeds labeled and unlabeled examples into a lower-dimensional latent space. LLP then propagates labels from known examples to unknown datapoints, weighting the likelihood of propagation by the local density of known examples. The neural network is then optimized to categorize all datapoints according to their pseudo-labels (with stronger emphasis on true known labels), while simultaneously encouraging datapoints sharing the same (pseudo-)labels to aggregate in the latent embedding space. The resulting embedding thus gathers both labeled images within the same class and unlabeled images sharing statistical similarities with the labeled ones. Through iteratively applying the propagation and network training steps, the LLP algorithm builds a good underlying representation for supporting downstream tasks, and trains an accurate classifier for the specific desired task.

We apply LLP in the context of object categorization in the ImageNet dataset (Deng et al., 2009), learning a high-performing network while discarding most of the labels. We show that LLP substantially outperforms previous scalable semi-supervised algorithms (Zhai et al., 2019; Miyato et al., 2018; Tarvainen & Valpola, 2017; Lee, 2013; Grandvalet & Bengio, 2005; Qiao et al., 2018) across a wide variety of training regimes, and that LLP-trained features support improved transfer to Places205, a large-scale scene-recognition task. We also present analyses that provide insights into the learning procedure and justification of key parameter choices.

## 2 RELATED WORK

Below we describe conceptual relationships between our work and recent related approaches, and identify relevant major alternatives for comparison.

**Deep Label Propagation.** Like LLP, Deep Label Propagation (Iscen et al., 2019) (DLP) also iterates between steps of label propagation and neural network optimization. In contrast to LLP, the DLP label propagation scheme is based on computing pairwise similarity matrices of learned visual features across all (unlabeled) examples. Unlike in LLP, the DLP loss function is simply classification with respect to pseudo-labels, without any additional aggregation terms ensuring that the pseudo-labeled and true-labeled points have similar statistical structure. The DLP method is effective on comparatively small datasets, such as CIFAR10 and Mini-ImageNet. However, DLP is challenging to apply to large-scale datasets such as ImageNet, since its label propagation method is $O(N^2)$ in the number $N$ of datapoints, and is not readily parallelizable. In contrast, LLP is $O(NM)$, where $M$ is the number of labeled datapoints, and is easily parallelized, making its effective complexity $O(NM/P)$, where $P$ is the number of parallel processes. In addition, DLP uniformly propagates labels across the embedding space, while LLP's use of local density-driven propagation weights specifically exploits the geometric structure in the space, improving pseudo-label inference.

**Deep Metric Transfer and Pseudolabels.** The Deep Metric Transfer (Liu et al., 2018) (DMT) and Pseudolabels (Lee, 2013) methods both use non-iterative two-stage procedures. In the first stage, the representation is initialized either with a self-supervised task such as non-parametric instance recognition (DMT), or via direct supervision on the known labels (Pseudolabels). In the second stage, pseudo-labels are obtained either by applying a label propagation algorithm (DMT) or naively from the pre-trained classifier (Pseudolabels), and these are then used to fine-tune the network. As in DLP, the label propagation algorithm used by DMT cannot be applied to large-scale datasets, and does not specifically exploit local statistical features of the learned representation. While more scalable, the Pseudolabels approach achieves comparatively poor results. A key point of contrast between LLP and the two-stage methods is that in LLP, the representation learning and label propagation processes interact via the iterative training process, an important driver of LLP's improvements.

**Self-Supervised Semi-Supervised Learning.** Self-Supervised Semi-Supervised Learning (Zhai et al., 2019) ($S^4L$) co-trains a network using self-supervised methods on unlabeled images and traditional classification loss on labeled images. Unlike LLP, $S^4L$ simply "copies" self-supervised learning tasks as parallel co-training loss branches. In contrast, LLP involves a nontrivial interaction between known and unknown labels via label propagation and the combination of categorization and aggregation losses, both factors that are important for improved performance.

**Consistency-based regularization.** Several recent semi-supervised methods rely on data-consistency regularizations. Virtual Adversarial Training (VAT) (Miyato et al., 2018) adds small input perturbations, requiring outputs to be robust to this perturbation. Mean Teacher (MT) (Tarvainen &

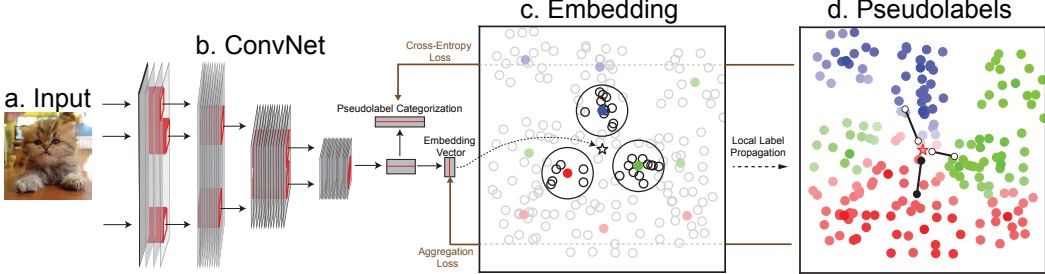

Figure 1: **Schematic of the Local Label Propagation (LLP) method. a.-b.** We use deep convolutional neural networks to simultaneously generate a lower-dimensional embedding and a category prediction for each input example. **c.** If the embedding of the input (denoted by ⋆) is unlabeled, we identify its close labeled neighbors (colored points), and infer ⋆'s pseudo-label by votes from these neighbors, with voting weights jointly determined by their distances from ⋆ and the density of their local neighborhoods (the highlighted circular areas). **d.** The pseudo-labels thereby created (colored points) come equipped with a confidence (color brightness), measuring how accurate the pseudo-label is likely to be. The network in **b.** is optimized (with per-example confidence weightings) so that its category predictions match the pseudo-labels, while its embedding is attracted (●—●) toward other embeddings sharing the same pseudo-labels and repelled (○—○) by embeddings of other pseudo-labels.

Valpola, 2017) requires the learned representation to be similar to its exponential moving average during training. Deep Co-Training (DCT) (Qiao et al., 2018) requires the outputs of two views of the same image to be similar, while ensuring outputs vary widely using adversarial pairs. Very recently, Unsupervised Data Augmentation (UDA) (Xie et al., 2019) achieves state-of-the-art performance by incorporating a substantially more complex data augmentation scheme into the data-consistency framework, and employing computationally expensive but practically impactful details such as the use of very large batch size during optimization. These methods all use unlabeled data in a "point-wise" fashion, applying the proposed consistency metric separately on each. They thus differ significantly from (and are thus likely complementary to) LLP, or indeed any method that explicitly relates unlabeled to labeled points. LLP benefits from training a shared embedding space that aggregates statistically similar unlabeled datapoints together with labeled (putative) counterparts. As a result, adding more unlabeled images consistently increases the performance of LLP, unlike for (e.g.) MT.

## 3 METHODS

We first give an overview of the LLP method. At a high level, LLP learns a model $f_\theta(\cdot)$ from labeled examples $X_L = \{x_1, \ldots, x_M\}$, their associated labels $Y_L = \{y_1, \ldots, y_M\}$, and unlabeled examples $X_U = \{x_{M+1}, \ldots, x_N\}$. $f_\theta(\cdot)$ is realized via a deep neural network whose parameters $\theta$ are network weights. For each input $x$, $f_\theta(x)$ generates two outputs (Fig. 1): an "embedding output", realized as a vector $v$ in a $D$-dimensional sphere, and a category prediction output $\hat{y}$. In learning $f_\theta(\cdot)$, the LLP procedure repeatedly alternates between two steps: *label propagation* and *representation learning*. First, known labels $Y_L$ are propagated from $X_L$ to $X_U$, creating pseudo-labels $Y_U = \{y_{M+1}, \ldots, y_N\}$. Then, network parameters $\theta$ are updated to minimize a loss function balancing category prediction accuracy evaluated on the $\hat{y}$ outputs, and a metric of statistical consistency evaluated on the $v$ outputs.

In addition to pseudo-labels, the label propagation step also generates [0,1]-valued *confidence scores* $c_i$ for each example $x_i$. For labeled points, confidence scores $C_L = \{c_1, \ldots, c_M\}$ are automatically set to 1, while for pseudo-labeled examples, confidence scores $C_U = \{c_{M+1}, c_{M+2}, ..., c_N\}$ are computed from the local geometric structure of the embedded points, reflecting how close the embedding vectors of the pseudo-labeled points are to those of their putative labeled counterparts. The confidence values are then used as loss weights during representation learning.

**Representation Learning.** Assume that datapoints $X = X_U \cup X_L$, labels and pseudo-labels $Y = Y_U \cup Y_L$, and confidences $C = C_U \cup C_L$ are given. Let $V = \{v_1, \ldots, v_N\}$ denote the set of corresponding embedded vectors, and $\hat{Y} = \{\hat{y}_1, \ldots, \hat{y}_N\}$ denote the set of corresponding category

prediction outputs. In the representation learning step, we update the network embedding parameters by simultaneously minimizing the standard cross-entropy loss $L_C(Y, \hat{Y})$ between predicted and propagated pseudo-labels, while maximizing a global aggregation metric $L_A(V|Y)$ to enforce overall consistency between known labels and pseudo-labels.

The definition of $L_A(V|Y)$ is based on the non-parametric softmax operation proposed in (Wu et al., 2018b;a). Specifically, we define the joint probability of any two embedding vectors $v_i$ and $v_j$ as:

$$P(v_i, v_j) = \frac{exp(v_i^T v_j / \tau)}{Z}, Z = \sum_{k=1}^{N} \sum_{l=1}^{N} exp(v_k^T v_l / \tau) \qquad (1)$$

where temperature $\tau \in (0, 1]$ is a fixed hyperparameter. Using this definition, we can get the probability of $v_i$ and the conditional probability of $v_j$ given $v_i$ as:

$$P(v_i) = \sum_{j=1}^{N} P(v_i, v_j) = \frac{\sum_{j=1}^{N} exp(v_i^T v_j / \tau)}{Z}, P(v_j|v_i) = \frac{P(v_i, v_j)}{P(v_i)} = \frac{exp(v_i^T v_j / \tau)}{\sum_{l=1}^{N} exp(v_i^T v_l / \tau)} \qquad (2)$$

Additionally, for $S \subset X$, its probability given $v_i$ is $P(S|v_i) = \sum_{v_j \in S} P(v_j|v_i)$. We then define the aggregation metric as the (negative) log likelihood of the examples whose pseudo-labels are also $y$, the pseudo-label of the current example $v$: $L_A(v) = -\log(P(A|v))$, where $A = \{x_i | y_i = y\}$. Optimizing $L_A(v)$ encourages the embedding corresponding to a given datapoint to selectively become close to embeddings of other datapoints with the same pseudo-label (Fig. 1).

The cross-entropy and aggregation loss terms are scaled on a per-example basis by the confidence score, and an $L2$ weight regularization penalty is added. Thus, the final loss for example $x$ is: $\mathcal{L}(x|\theta) = c \cdot [L_C(y, \hat{y}) + L_A(v)] + \lambda \|\theta\|_2^2$, where $\lambda$ is a regularization hyperparameter.

**Label Propagation.** To understand how LLP generates pseudo-labels $Y_U$ and confidence scores $C_U$, it is useful to start from the weighted K-Nearest-Neighbor classification algorithm (Wu et al., 2018b), in which a "vote" is obtained from the top $K$ nearest labeled examples for each unlabeled example $x$, denoted $\mathcal{N}_K(x)$. The vote of each $x_i \in \mathcal{N}_K(x)$ is weighted by the corresponding probabilities $P(v_i|v)$. Assuming $S$ classes, the total weight for pseudo-labeled $v$ as class $j$ is thus:

$$w_j(v) = \sum_{i \in I^{(j)}} P(v_i|v), \quad \text{where} \quad I^{(j)} = \{i | x_i \in \mathcal{N}_K(v), y_i = j\} \qquad (3)$$

Therefore, the probability $p_j(v)$ that datapoint $x$ is of class $j$, the associated inferred pseudo-label $y$, and the corresponding confidence $c$, may be defined as: $p_j(v) = w_j(v) / \sum_{k=1}^{S} w_k(v)$, $y = \arg\max_j p_j(v)$, and $c = p_y(v)$. Although intuitive, weighted-KNN introduces a positive correlation between the local density of a labeled example and the number of unlabeled examples whose pseudo-labels are propagated from this labeled example. Moreover, if the labeled examples within one category have higher densities than other categories, there will be more unlabeled examples pseudolabled as this category. To avoid this correlation, we additionally penalize the labeled examples with higher densities through dividing the KNN weight of a labeled example by its local density. To formalize this penalization, we divide $P(v_i|v)$ in the definition of $w_j(v)$ with a local density-weighted probability:

$$P^L(v_i|v) = P(v_i|v)/\rho(v_i) \text{ where } \rho(v_i) = \sum_{j \in \mathcal{N}_T(v_i)} P(v_j, v_i) \qquad (4)$$

where $\mathcal{N}_T(v_i)$ are $T$ nearest unlabeled examples and denominator $\rho(v_i)$ is a measure of the local embedding density. For consistency, we replace $\mathcal{N}_K(v)$ with $\mathcal{N}_K^L(v)$, which contains the $K$ labeled neighbors with highest locally-weighted probability, to ensure that the votes come from the most relevant labeled examples. The final form of the LLP propagation weight equation is thus:

$$w_j(v) = \sum_{i \in I^{(j)}} \frac{P(v_i|v)}{\sum_{k \in \mathcal{N}_T(v_i)} P(v_k, v_i)}, \text{where } I^{(j)} = \{i | i \in \mathcal{N}_K^L(v), y_i = j\} \qquad (5)$$

The intuition behind the local density weighting idea is further quantitatively explored in §5.

**Memory Bank.** Both described steps implicitly require access to all the embedded vectors $V$ at every step. However, recomputing $V$ becomes intractable for bigger dataset. We address this issue by approximating realtime $V$ with a memory bank $\bar{V}$ that keeps a running average of the embeddings. As this procedure is taken from Wu et al. (2018b), we refer readers there for a detailed description.

Table 1: Top-1 accuracy (%) of ResNet-18 on ImageNet with varying $p$ and $q = 100$

| Method | $p = 1$ | $p = 3$ | $p = 5$ | $p = 10$ |
|---|---|---|---|---|
| Supervised | 17.35 | 28.61 | 36.01 | 47.89 |
| DCT (Qiao et al., 2018) | – | – | – | 53.50 |
| MT (Tarvainen & Valpola, 2017) | 16.91 | 40.81 | 48.34 | 56.70 |
| LLP (ours) | 27.14 | 53.24 | 57.04 | 61.51 |
| LLP + RJ (ours) | **33.55** | **55.30** | **58.92** | **63.20** |

Table 2: Top-5 accuracy (%) of ResNet-50 on ImageNet with $p = \{1, 10\}$ and $q = 100$

| $p$ | Supervised | Pseudolabels | VAT-EM | $S^4L^1$ | MT | LLP (ours) | LLP + RJ (ours) |
|---|---|---|---|---|---|---|---|
| 1 | 48.43 | 51.56 | 46.96 | 53.37 | 40.54 | 61.89 | **72.20** |
| 10 | 80.43 | 82.41 | 83.39 | 83.82 | 85.42 | 88.53 | **89.55** |

## 4 RESULTS

We first evaluate the LLP method on visual object categorization in the large-scale ImageNet dataset (Deng et al., 2009), under a variety of training regimes. We also illustrate transfer learning to Places 205 (Zhou et al., 2014), a large-scale scene-recognition dataset.

**Experimental settings.** We follow Wu et al. (2018b) for most of our hyperparameters and optimization settings. In the label propagation step, we set $K = 10$ and $T = 25$ (these choices are justified in Section 5). We use ResNet-18v2 and ResNet-50v2 (He et al., 2016b). We find a "rate-jump" phase beneficial: after the initial schedule for the learning rate, increasing and dropping it again improves the performance. We think this is because a larger learning rate is needed to leverage the better embedding space in the later training, especially given that the performance usually has big jumps after dropping the learning rate and the dropped learning rate is already too small to exploit the improved quality. To clearly show the effect of this phase, we list the performance after applying it using "LLP + RJ" in Table 1-2. We also apply this phase to supervised learning and Mean Teacher and their performances are not changed. More details are in Appendix A. We train on ImageNet with $p\%$ labels and $q\%$ total images available, meaning that $M \sim p\% \times 1.2M$, $N \sim q\% \times 1.2M$. Different regimes are defined by $p \in \{1, 3, 5, 10\}$ and $q \in \{30, 70, 100\}$. Results are shown in Tables 1-3. Due to the inconsistency of reporting metrics across different papers, we alternate between comparing top1 and top5, depending on which was reported in the relevant previous work.

The results show that: **1.** LLP significantly outperforms previous state-of-the-art methods by large margins within all training regimes tested, regardless of network architectures, $p$, and $q$. **2.** When compared to the very recent UDA approach (Xie et al., 2019) for ResNet-50 with $p = 10$ and $q = 100$, LLP achieves better top5 (89.55 v.s. 88.52, LLP v.s. UDA) and top1 (70.85 v.s. 68.66). Moreover, LLP has the potential to achieve even higher performance if trained with the complex preprocessing pipeline and large batch size used in UDA (15360 for UDA vs 64 for LLP here, chosen due to computational resource limitations), as these details have been shown to meaningfully improve performance; **3.** LLP shows especially large improvements to other methods when $p$ is small. For example, ResNet-18 trained using LLP with $p = 1$ surpasses MT by **16.64%** in top1 and our ResNet-50 with $p = 1$ surpasses $S^4$L by **18.83** in top5 (UDA does not report for less than 10% labels).

**Leveraging additional unlabeled images.** To examine how good LLP is at using unlabeled images, we first vary the value of $q$ while $p$ remains at 10. As shown in Table 3, LLP consistently benefits from additional unlabelled images and is not yet saturated using the whole ImageNet, unlike Mean Teacher, where the number of unlabelled images appears essentially irrelevant.

To further assess how LLP might behave in noisier real-world settings, we additionally performed a preliminary exploration using the YFCC100M (Thomee et al., 2015) dataset as a source of augmenta-

---

[1] For $S^4L$, we list their $S^4L$-Rotation performance, which is their best reported performance using ResNet-50. Note that although a model with higher performance is reported by $S^4L$, that model uses a much more complex architecture than ResNet-50.

Table 3: Top-1 accuracy (%) of ResNet-18 on ImageNet with $p = 10$ and varying $p$. "FT" means the fine-tuning process used in YFCC100M experiments, which is the same as the "rate-jump" phase.

| Method | ImageNet | | | | ImageNet + YFCC100M | |
| | $q = 30$ | $q = 70$ | $q = 100$ | $q = 100+$ FT | $q = 100+$**FAR** | $q = 100+$**NEAR** |
|---|---|---|---|---|---|---|
| MT | 56.07 | 55.59 | 55.65 | - | - | - |
| LLP (ours) | **58.62** | **60.27** | **61.51** | 63.20 | 63.14 | 63.53 |

Table 4: ResNet-50 transfer learning Top-1 accuracy (%) on Places205 using weights pretrained on ImageNet with $p = \{1, 10\}$. Most numbers are from Zhai et al. (2019). *: Produced by us.

| # labels | Supervised | Pseudolabels | VAT | VAT-EM | $S^4L$ | MT | LLP (ours) | LA* |
|---|---|---|---|---|---|---|---|---|
| 10% | 44.7 | 48.2 | 45.8 | 46.2 | 46.6 | 46.4 | **50.4** | 48.3 |
| 1% | 36.2 | 41.8 | 35.9 | 36.4 | 38.0 | 31.6 | **48.5** | |

tion. However, because YFCC100M is drawn from a very different distribution than ImageNet, we select a subset of images that are more similar to ImageNet using the pipeline proposed by (Yalniz et al., 2019). This pipeline is applied to two randomly chosen subsets of YFCC100M with 5M and 10M images, respectively, creating two selected training subsets each of roughly 480K images — denoted **FAR** and **NEAR** — differing in how close to the ImageNet distribution the sets are. We then combine each selected set with ImagetNet of $p = 10$ and $q = 100$ and train LLP. After this training, we finetuned the model with only ImagetNet data using LLP, following the procedure in (Yalniz et al., 2019). Please refer to Appendix B for more details of the selection and fine-tuning processes.

The results in Table 3 show that even with this very preliminary attempt, LLP achieves a 2.02% performance improvement with augmentation of images chosen from the **NEAR** augmentation, compared to the 61.51% baseline (though the fine-tuning process likely accounts for part of this improvement). Almost certainly, such augmentations would be substantially greater if a larger number of images from a better-matched distribution were available and a better network is used for selecting the images (Yalniz et al., 2019). Unfortunately, a direct comparison between LLP and Yalniz et al. (2019) is not presently possible as their ResNet18 result uses the entire labeled ImageNet and all of YFCC100M to select matched augmentation images, both of which require significantly more computation resources than are available to us. However, it is worth noting that LLP benefits even from selecting the matched dataset from 10M YFCC images, while Yalniz et al. (2019) needs more than 20M images to achieve a similar gain.

**Transfer learning to Scene Recognition.** To evaluate the quality of our learned representation in other downstream tasks besides ImageNet classification, we assess its transfer learning performance to the Places205 (Zhou et al., 2014) dataset. This dataset has $2.45M$ images total in 205 distinct scene categories. We fix the nonlinear weights learned on ImageNet, add another linear readout layer on top of the penultimate layer, and train the readout using cross-entropy loss using SGD as above. Please refer to Appendix C for other details. We only evaluate our ResNet-50 trained with $p = \{1, 10\}$, as Zhai et al. (2019) reported performance with this setting. Table 4 show that LLP again significantly outperforms previous state-of-the-art results. It is notable that when trained with $p = 1$, only LLP shows slightly better performance to Places205 than the Local Aggregation (LA) method, the current state-of-the-art unsupervised learning method (Zhuang et al., 2019).

## 5 ANALYSES

**Emerging clusters during training.** Intuitively, the aggregation term $L_A(v)$ should cause embedding outputs with the same label, whether known or propagated, to cluster together during training. Fig. 2a shows clustering becoming more pronounced along the training trajectory both for labelled and unlabelled datapoints, while unlabelled datapoints surround labelled datapoints increasingly densely. A simple metric measuring the aggregation of a group of embedding vectors is the L2 norm of the group mean, which, since all embeddings lie in the 128-D unit sphere, is inversely related to the group dispersion. Computing this metric for each category and averaging across categories, we obtain a quantitative description of aggregation over the learning timecourse (Fig. 2b), further supporting the conclusion that LLP embeddings become increasingly clustered. We also investigate how network

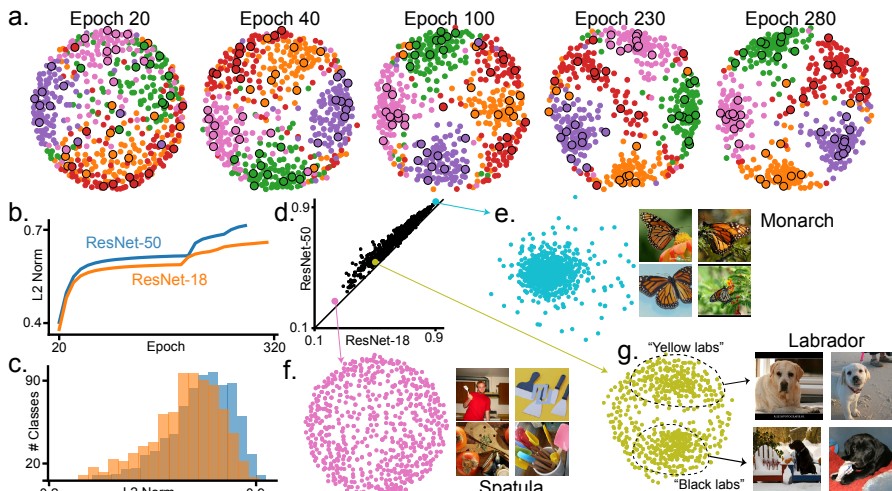

Figure 2: **a.** MDS embeddings of 128-D embedding outputs on 100 random images from each of five random ImageNet categories, from the beginning to the end of LLP training. Larger points with black borders are images with known labels. **b.** Trajectory of cross-category average of L2-norms of category-mean embedding vectors. Sudden changes are due to learning rate drops. **c.** Histogram of the L2-norm metrics for each category, for fully-trained ResNet-18 and ResNet-50 networks. **d.** Scatter plot of L2-norm metric for ResNet-18 ($x$-axis) and ResNet-50 ($y$-axis). Each dot represents one category. **e.-g.** MDS embeddings and exemplar images for images of "Monarch", "Spatula", and "Labrador". For each category, MDS embedding is computed from 700 random images.

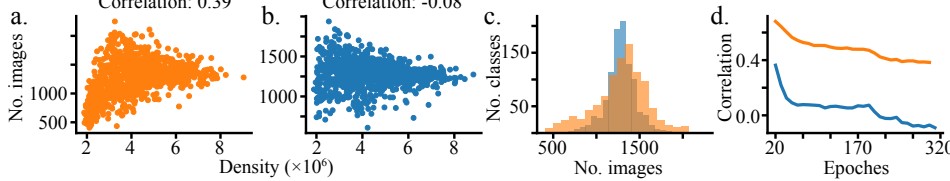

Figure 3: Density weighting analysis. Orange color is for "NoDW" model and blue color is for LLP model. **a, b.** Scatter plot of ImageNet classes. Each dot represents one class. For class $i$, X-axis is $D_i$ and Y-axis is $Q_i$. Both models are fully trained. **c.** Histogram plot of classes showing the distribution of $Q_i$. **d.** The trajectory of Pearson correlation between $\{D_i\}$ and $\{Q_i\}$ during training.

architecture influences learning trajectory and the final representation, comparing ResNet-50 and ResNet-18 trained with 10% labels (Fig. 2b-d). The more powerful ResNet-50 achieves a more clustered representation than ResNet-18, across timepoints and categories.

**Category structure analysis: successes, failures, and sub-category discovery.** It is instructive to systematically analyze statistical patterns on a per-category basis. To do this, we visualize the embeddings for three representative categories with 2D multi-dimensional scaling (MDS). For an "easy" category with a high aggregation score (Fig. 2e), the LLP embedding identifies images with strong semantic similarity, supporting successful image retrieval. For a "hard" category with low score (Fig. 2f), image statistics vary much more and the embedding fails to properly cluster examples together. Most interestingly, for multi-modal categories with intermediate scores (Fig. 2g), the embedding can reconstruct semantically meaningful sub-clusters even when these are not present in the labelling e.g. the "labrador" category decomposing into "black" and "yellow" subcategories.

**Comparison to global propagation in the small-dataset regime.** To understand how LLP compares to methods that use global similarity information, but therefore lack scalability to large datasets, we test several such methods on ImageNet subsets (see Appendix D for details). Table 5 shows that LLP can be effective even in this regime, as it is comparable to the global propagation algorithm used in DMT (Liu et al., 2018) and only slightly lower than DLP (Iscen et al., 2019).

**Ablation studies.** To illustrate the importance of key design choices in LLP, we conduct a series of ablation studies exploring the following alternatives, using: 1. Different $K$s (experiments **Top50**,

Table 5: Label propagation performance on ImageNet subsets. LS: Label Spreading (Zhou et al., 2004). LP: Label Propagation (Zhu & Ghahramani, 2002). LP_DMT: the method in DMT (Liu et al., 2018). LP_DLP: the method in DLP (Iscen et al., 2019). Standard deviations are across 10 subsets.

| Method | LS | LP | LP_DMT | LP_DLP | LLP (ours) |
|--------|-----|-----|--------|--------|-----------|
| Perf. | $84.6 \pm 3.4$ | $87.7 \pm 2.2$ | $88.2 \pm 2.3$ | $89.2 \pm 2.4$ | $88.1 \pm 2.3$ |

Table 6: Top1 accuracy (%) for ResNet-18 and ResNet-50 trained with $p = 10, q = 100$ and other settings. "TopX" means training with $K$ =X. "NoC" does not weight the loss by confidence. "NoDW" means the KNN weight is not weighted by densities.

| Model | Top50 | Top20 | Top5 | NoC | NoDW | LLP |
|-------|-------|-------|------|-----|------|-----|
| Res18 Perf. | 61.07 | 61.48 | 60.90 | 55.88 | 58.41 | **61.51** |
| Res50 Perf. | – | – | – | 64.01 | 66.41 | **68.90** |

**Top20**, and **Top5** in Table 6); 2. Confidence weighting, or not (**NoC**); 3. Density-weighted probability, or not (**NoDW**). Table 6 shows the contributions of each design choice, indicating that both confidence weighting and density weighting lead to significant performance gains, across architectures.

**Understanding density weighting.** To better explain why the density weighting method is useful, we compute two measures for each class $i$: the average density of its labeled examples, denoted $D_i$; and the number of unlabeled examples pseudolabled as $i$, denoted $Q_i$. Formally, these are defined by: $D_i = \sum_{j \in L_i} Z\rho(v_j)/\|L_i\|$, where $L_i = \{j | x_j \in X_L, y_j = i\}$, and $Q_i = \|\{j | x_j \in X_U, y_j = i\}\|$. Fig. 3a illustrates the strong positive correlation between $D_i$ and $Q_i$ for the unweighted **NoDW** model, which leads to an imbalanced distribution of $Q_i$ shown in Fig. 3c. After applying density weighting, $D_i$ and $Q_i$ become decorrelated (Fig. 3b), creating an empirically accurate balanced pseudo-label class distribution, throughout optimization (Fig. 3d). Another potential method to enforce an empirically correct $Q_i$ distribution would be to reweight KNN coefficients to directly reflect the empirical label ratio, replacing $P^L(v_i|v)$ in eq. 4 with $P^R(v_i|v) = P(v_i|v) \times \frac{L_{y_i}}{M} / \frac{Q_{y_i}}{N-M}$. However, this simple "ratio-based" scheme does not explicitly address the local correlation of $Q_i$ and $D_i$. Indeed, an experiment with $P^R(v_i|v)$ in place of $P^L(v_i|v)$ using ResNet-18 on 10% labeled ImageNet only achieves top1 59.4%, substantially worse than LLP, further supporting the effectiveness of the more sophisticated local density weighting approach.

## 6 CONCLUSION

In this work, we presented LLP, a method for semi-supervised deep neural network training that efficiently propagates labels from known to unknown examples in a common embedding space, ensuring high-quality propagation by exploiting the local structure of the embedding. The embedding itself is simultaneously co-trained to achieve high categorization performance while enforcing statistical consistency between real and pseudo-labels. LLP achieves state-of-the-art semi-supervised learning results across all tested training regimes, including those with very small amounts of labelled data, and transfers effectively to other non-trained tasks.

In future work, we seek to improve LLP by better integrating it with state-of-the-art unsupervised learning methods (e.g. Zhuang et al. (2019)). This is especially relevant in the regime with very-low fractions of known labelled datapoints (e.g. <1% of ImageNet labels), where the best unsupervised methods outperform state-of-the-art semi-supervised methods. Combining LLP with the very distinct point-wise methods in MT or UDA is also of interest, as would be the effective use of larger computational resources to enable conceptually simple but practically important optimization details such as (e.g.) significantly larger batch size in UDA. In addition, in its current formulation, LLP may be less effective on small datasets than alternatives that exploit global similarity structure (e.g. Iscen et al. (2019); Liu et al. (2018)). We thus hope to improve upon LLP by identifying methods of label propagation that can take advantage of global structure while remaining scalable.

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

## A    IMAGENET EXPERIMENT DETAILS

Following Wu et al. (2018b); Zhuang et al. (2019), $\tau = 0.07$ and $D = 128$. Optimization uses SGD with momentum of 0.9, batch size of 128, and weight-decay parameter $\lambda = 0.0001$. Learning rate is initialized to 0.03 and then dropped by a factor of 10 whenever validation performance saturates. Depending on the training regime specifics (how many labelled and unlabelled examples), training takes 200-400 epochs, comprising three learning rate drops. After that, the learning rate is increased to 0.003 and dropped for two more times as a "relearning" phase. This additional phase can take around 200 more epochs. During training, pseudolabels are updated every step for the current examples after the gradient updates. The density estimate $\rho(v_i)$ is recomputed for all labelled images at once at the end of every epoch. For the network architectures, we add an additional fully connected layer alongside the standard softmax categorization layer to generate the embedding output. Although the aggregation loss requires the computation of $P(v)$, which involves getting the dot product results of $v$ and all vectors stored in memory bank, we find this operation can be efficiently executed through GPUs, which makes the noise-contrastive estimation in Wu et al. (2018b) unnecessary. Our algorithm only adds a little extra computation time compared to a purely supervised training regime. As a reference, a ResNet-18v2 trained on two Titan-Xps requires around 10 days to be fully-trained after 400 epochs. This time can be further reduced to less than one week through using NVIDIA DALI preprocessing library. We use DALI for all of our main experiments except those with $p = 1$.

## B    IMAGENET + YFCC100M EXPERIMENT DETAILS

To select images that are more similar to ImageNet, we apply a LLP trained ResNet-18 model on ImageNet with 10% labels to each image, predict its class confidence score, and retain $P$ classes with highest scores for this image. Then for each class, we select the top $K$ images by confidence. Following Yalniz et al. (2019), we set $P = 10$. $K$ is heuristically set to be 500, based on the explorations done in Yalniz et al. (2019). A more thorough parameter search on $P$ and $K$ may lead to even better results than reported here.

For the fine-tuning process, we take the fully trained models and restart the LLP training with learning rate 0.003. We then keep the training for two more learning drops.

## C    PLACES205 EXPERIMENT DETAILS

Learning rate is initialized at 0.01 and dropped by factor of 10 whenever validation performance on Places205 saturates. Training requires approximately 500,000 steps, comprising two learning rate drops.

## D    PROPAGATION IN THE SMALL-DATASET REGIME

ImageNet subsets are constructed through randomly sampling 50 categories from ImageNet and 50 images from each category. For each category selected, we choose 5 images to be labelled. For all methods, we use embedding outputs of our trained ResNet-50 with $p = 10$ as data features.

