# OpenReview forum: "Local Label Propagation for Large-Scale Semi-Supervised Learning"
_ICLR.cc/2020/Conference — Reject_

### Official Review · AnonReviewer1 · 2019-10-22
**Official Blind Review #1**

**Rating:** 6

**Review:**

The paper introduces an approach for semi-supervised learning based on local label propagation. The idea is to leverage the geometric structure in the embedding space, such that data near to each other in the embedding space should have the same labels. The labels of the K-nearest labeled examples are weighted to form the propagated pseudo label of the data point. And the objective aims to match the propagated pseudo label and the predicted label from the classification model. An extra term is added to the objective to force data points with similar pseudo labels to get close to each other in the embedding space. The local propagation strategy makes the method scalable compared to similar methods in the literature. The method is tested on different experimental setups and show superior performance than the state of the art baselines.

I like the idea of learning a consistent embedding space for prediction and label propagation seems interesting and novel to my knowledge. However, the paper uses the technique of local aggregation in [2] to replace the label propagation in [1], which makes it less novel.

Also, the experiments seem to be extensive with additional analysis to the behavior of the proposed method. I noticed that the authors used the unsupervised training proposed in [2] for the first 10 epochs. I wonder how important it is to have this initialization and would like to see ablation studies on whether using this as initialization or not.

[1]Iscen, Ahmet, Giorgos Tolias, Yannis Avrithis, and Ondrej Chum. "Label propagation for deep semi-supervised learning." In Proceedings of the IEEE Conference on Computer Vision and Pattern Recognition, pp. 5070-5079. 2019
[2]Zhuang, Chengxu, Alex Lin Zhai, and Daniel Yamins. "Local aggregation for unsupervised learning of visual embeddings." arXiv preprint arXiv:1903.12355 (2019).

**Experience Assessment:**

I do not know much about this area.

**Review Assessment: Checking Correctness Of Derivations And Theory:**

I carefully checked the derivations and theory.

**Review Assessment: Checking Correctness Of Experiments:**

I assessed the sensibility of the experiments.

**Review Assessment: Thoroughness In Paper Reading:**

I read the paper thoroughly.

---

> ### Author Response · Authors · 2019-11-12
> **Response to your review**
>
> "I like the idea of learning a consistent embedding space for prediction and label propagation seems interesting and novel to my knowledge."
>
> Thanks!
>
> "However, the paper uses the technique of local aggregation in [2] to replace the label propagation in [1], which makes it less novel."
>
> Yeah, this is definitely a natural potential area for confusion! Despite similar-ish names and use of an embedding-based framework, LLP is actually fairly different from the idea of "using Local Aggregation (LA) to replace the label propagation." There are two key differences between what LLP does and what that simpler idea (but natural) would involve:
>
> 1. The loss function in LLP is quite different from that in LA.  LA identifies "close" and "background" neighbors for each example, then aggregates that example toward close neighbors while separating it from the background. In LLP, by contrast, there is no process that identifies two sets of neighbors, nor are clustering techniques used at all. Instead, LLP first finds nearest labeled examples for one unlabeled example in the embedding space and then weights their labels by both the distance in the embedding space, and also the local densities of these labeled examples to infer the pseudolabel for that unlabeled example. This procedure is just mathematically very different from LA, with both a different loss functional form and the involvement of different basic operations (local density and confidence reweighting vs clustering).
>
> 2. The label propagation algorithm used by LLP itself is quite different to the algorithm proposed in [1]. Most importantly, LLP formulates a novel label propagation algorithm via distances in the embedding space, using mathematically well-defined probability measures (the "non-parametric softmax" concept). In addition, the loss optimized by LLP includes both the categorization and the aggregation losses, while Iscen et al. only uses categorization loss.
>
> So these two differences end up at a state where LLP is ultimately fairly different to "replace label prop with LA". We realize we need to explain the differences between LA and LLP better, and will make sure to do this in the revised version of the paper.
>
> (The following comment is a just a semi-related thought on the values of quality in machine learning.  Feel free to ignore it (but we hope at least someone is reading this!))
> From one point of view, the fact that LLP is not just "LA as label prop" is "good" since it makes LLP "more novel." But in another sense, we think the need for this novelty is actually a downside. In our view, it would have been better scientifically and conceptually had a very simple scheme using LA as standard label prop actually been a workable solution. We would gladly have presented that instead, even if that meant the contribution were less "novel".  That's because, had that been the case, there would have been a more unified solution to the related but different problems of unsupervised and semi-supervised learning. We think the search for "novelty" is probably somewhat distorting as a scientific value, because sometimes (perhaps oftentimes) the simplest solution is better than a complicated solution, and over-valuing novelty sets up a bit of a bias against simple solutions.  Ultimately, we think it would be better for the community if carefulness of evaluation, and strength of empirical results, were a bit more highly valued. \endofscreed
>
> "I noticed that the authors used the unsupervised training proposed in [2] for the first 10 epochs. I wonder how important it is to have this initialization and would like to see ablation studies on whether using this as initialization or not."
>
> Thanks for asking this. We have just tested the experiment starting from scratch and its performance seems the same as starting from pretrained models by unsupervised learning algorithms. We will update our paper about this.

---

### Official Review · AnonReviewer3 · 2019-10-25
**Official Blind Review #3**

**Rating:** 3

**Review:**

The authors propose a local label propagation approach for large-scale semi-supervised learning. The approach learns a representation that tries to minimize a combination of the cross-entropy loss on the labeled data and a negative inner-product-based likelihood between the propagated pseudo-label and other examples with the same true label. The pseudo-labels on the unlabeled data are then calculated with a weighted k-NN scheme, where the weights take a heuristic correction of a soft similarity. Some further computational speedup is done with a memory cache described in an earlier work (Wu 2018b). Experimental results seem significantly superior to the competitors. The design choices are mostly justified with ablation studies.

The whole idea is interesting and the results are promising. From the current manuscript, my remaining concerns are

(1) How much contribution has readily been done by (Wu 2018a, b), and how much is the original design of the authors? From Section 3 (and without reading (Wu 2018a, b)), I cannot find a clear answer to this question. Currently it appears that the additional contribution over Wu's works is marginal.

(2) It is not clear to me how the proposed approach reaches the asserted efficiency over global label propagation approaches. In particular, each P(v_i)*Z in Equation (2) is O(N) to compute. Each w_j(v) in Equation (5) is O(K) to compute after getting all P(v_i)*Z, and then there are N (or at least N-M) such w_j(v) needed. So the total complexity is naively O(N (N-M) K). Even ignoring the K as a small constant, I cannot see how LLP is O(NM). Some running time profiling of LLP versus global LP might be helpful.

(3) For label propagation methods, it is important to understand whether the pseudo-labels are accurate and/or whether the methods might be mis-guided by the pseudo-labels. Is there any evidence on whether the pseudo-labels are accurate (absolutely, or with respect to the confidence)?

(4) For hyper-parameter selection, there is a "Learning rate is initialized to 0.03 and then dropped by a factor of 10 whenever validation performance saturates." But it is not clear how the validation set is formed, and what performance is measured. Is it a performance based on a labeled validation set (and if so, how large is the set) or unlabeled one?

I read the rebuttal. While it does not change my assessment, I thank the authors for clarifying some issues.

**Experience Assessment:**

I do not know much about this area.

**Review Assessment: Checking Correctness Of Derivations And Theory:**

I assessed the sensibility of the derivations and theory.

**Review Assessment: Checking Correctness Of Experiments:**

I assessed the sensibility of the experiments.

**Review Assessment: Thoroughness In Paper Reading:**

I read the paper at least twice and used my best judgement in assessing the paper.

---

> ### Author Response · Authors · 2019-11-10
> **Response to your first concern**
>
> "... The whole idea is interesting and the results are promising. ..."
> Thanks!
>
> "(1) How much contribution has readily been done by (Wu 2018a, b), and how much is the original design of the authors? ... Currently it appears that the additional contribution over Wu's works is marginal."
>
> Although we situate our work in the "non-parametric softmax" embedding framework introduced in the Wu et. al papers, our work solves a really different research problem from those works, and has to use a very different core algorithm to do so. At a high level, the key difference in research problem is that Wu 2018a is for supervised learning using deep embeddings, while Wu 2018b is for unsupervised learning, e.g. learning from only unlabeled data. In contrast, our work is for semi-supervised learning, which involves learning from the \emph{interaction} between labelled and un-labelled data. To perform semi-supervised learning well means leveraging this interaction in a very non-trivial way, and doing so requires a really different algorithmic approach than needed for either supervised or unsupervised learning alone. Thus, the core elements of LLP's technical approach end up having to be different from that in (Wu 2018a, b). The most striking difference, of course, is that LLP involves a label propagation process, which is a very different type of operation from that used in the Wu et al work.  The whole idea of label-propagation-based inference of pseudolabels just doesn't exist in the Wu papers (which is understandable, since those papers solved a very different problem). After running label propagation, LLP then uses a novel local aggregation approach to establish consistency between the pseudolabels and real labels, leading to really very different loss function than in the Wu work.  LLP then iterates between label prop and learning steps to generate the final representation. Neither Wu 2018a or Wu 2018b has a label propagation procedure, a local aggregative loss optimization, or an iterative learning procedure. Moreover, our work also introduces a very novel local-density-weighting method for the KNN propagation algorithm, which as shown in the ablation studies significantly improves the performance. This method comes from our insights about the resulted embedding space and resolves a specific and important problem in semi-supervised learning in this embedding space (see the last ablation study in the paper for more details). So, in summary the fact that LLP is situated in the deep embedding framework is a useful mathematical convenience, but the embedding goal, and the mathematical technique used to achieve it, is really very different for LLP than in the Wu et al work.

---

> ### Author Response · Authors · 2019-11-10
> **Response to your second concern**
>
> "(2) It is not clear to me how the proposed approach reaches the asserted efficiency over global label propagation approaches. In particular, each P(v\_i)*Z in Equation (2) is O(N) to compute. Each w\_j(v) in Equation (5) is O(K) to compute after getting all P(v\_i)*Z, and then there are N (or at least N-M) such w\_j(v) needed. So the total complexity is naively O(N (N-M) K). Even ignoring the K as a small constant, I cannot see how LLP is O(NM). Some running time profiling of LLP versus global LP might be helpful."
>
> This is a good question, and an important point since scalability is one of the key motivations of our work.  Two points that are actually important for the efficiency were not well emphasized, which we'll explain here now, and also update the revised paper to explain better. The first point is that although computing $w_j(v)$ requires $P(v_i)\times Z$, it is $p_j(v)$ that is actually used in the loss function and as $p_j(v) = w_j(v) / \sum_{k=1}^S w_k(v)$, $P(v_i)\times Z$ is actually cancelled out, and no longer needed.  More specifically, $p_j(v) = \frac {\sum_{i \in I^{j}} exp(v_i^Tv / \tau) / \rho(v_i)} {\sum_{k=1}^M exp(v_k^Tv / \tau) / \rho(v_k)}$.  This is very important, since if we had to compute $P(v_i)\times Z$ explicitly, that would probably be quite expensive. The second point is that pseudolabels are propagated from labeled points to unlabeled points. Once we compute local densities, then the computation of all $p_j(v)$ for one unlabeled example only requires dot product results between labeled examples and the current example, which is $O(M)$. Doing this for all unlabeled points will then take $O((N-M)M)$ --- which, after all, is still linear in $N$. As for computing the local densities of labeled examples in the first place, this only needs to be done infrequently (we get away with doing it only once each epoch).  Moreover, it only takes about 40s to compute the local densities for 120K labeled examples with 1.1M unlabeled examples, so this is a pretty minor additional overhead. And, label propagation itself is run online during training for each example in the current batch and this only adds very little time to the training. In contrast to the efficiency of LLP, if we apply global label propagation such as Label Propagation method from Zhu \& Ghahramani, 2002, it would take several hours per batch, even when implemented on gpus. This is why a method such as ours is necessary -- e.g. being able to construct a nontrivial procedure that takes advantage of the interaction between labelled and non-labelled examples but still runs with the efficiency as if it were essentially regular deep learning, is a really key step. We'd be happy to add this more detailed time complexity analysis into our paper.

---

> ### Author Response · Authors · 2019-11-10
> **Response to your third and forth concerns**
>
> "(3) For label propagation methods, it is important to understand whether the pseudo-labels are accurate and/or whether the methods might be mis-guided by the pseudo-labels. Is there any evidence on whether the pseudo-labels are accurate (absolutely, or with respect to the confidence)?"
>
> We should have mentioned in our paper that the accuracy of our pseudolabels on the training set is usually close to but slightly lower than the performance on the validation set. For example, the pseudolabels of a ResNet18 trained on ImageNet with 10% labeled images are 59.61% correct. In thinking about your question a little further, we also found that there is a strong correlation between the LLP algorithms estimated confidence of the pseudolabels and their actual correctness. For example, the pseudolabels of the images whose confidence is between 0.1 and 0.2 are 5.37% correct by average, while for confidence between 0.9 and 1, the average accuracy is 90.81%.  This is presumably \emph{why} confidence weighting is so useful.  We will add a figure showing this effect in our revision.  Thanks for spurring us to do this analysis!
>
> "(4) For hyper-parameter selection, there is a "Learning rate is ... dropped by a factor of 10 whenever validation performance saturates." But it is not clear how the validation set is formed, and what performance is measured. Is it a performance based on a labeled validation set (and if so, how large is the set) or unlabeled one?"
>
> We use the standard public ImageNet validation set (provided with the ILSVRC 2012 challenge) as our validation set. This set includes 50,000 images in total, as each of the 1000 categories has 50 images included. Because the validation set is so large, it has been shown that validation set performance is very highly correlated with performance on the held-out (non-public) final test set that was used for the ImageNet challenges.  This validation set is totally non-overlapping with the imageset we use during training, which is standard ImageNet ILSVRC 2012 training set (with ~1.2M images).  We report performance either on top1 or top5 categorization performance (as appropriate for comparison purposes), as measured on the validation set (not the training set). Overall, we have followed what is essentially the "standard procedure" in the field for using ImageNet, and which is identical to that used by the works that to which we compare our results.

---

### Official Review · AnonReviewer2 · 2019-10-26
**Official Blind Review #2**

**Rating:** 6

**Review:**

The paper discusses a new strategy for deep semi-supervised learning that seems related to the deep label propagation method of Iscen et al. 2019, but is more scalable and has a different loss function.

Each example is associated with a representation vector v_i and a label y_i. The authors' approach essentially works by alternating between two steps:

(1) Representation learning: Updating the v_i (and other model parameters) where the loss is an addition of two terms:
-standard supervised loss
-term that encourages points with similar labels to have similar v_i

(2) Label Propagation: Uses the representations v_i to compute nearest neighbors and propagate labels. The authors approach takes O(NM) where N is total number of points and M is number of labeled points and can be parallelized to O(NM/P).  This is in contrast to Iscen et al. 2019 which takes O(N^2).

Experiments show that the authors' approach performs consistently better on ImageNet than existing approaches. With suboptimal preprocessing, it also performs comparable / slightly better than UDA (Xie et al. 2019) (The authors speculate it could do better with the preprocessing that UDA uses)

I am not from this area but found the paper well written and easy to understand.


**Experience Assessment:**

I do not know much about this area.

**Review Assessment: Checking Correctness Of Derivations And Theory:**

I assessed the sensibility of the derivations and theory.

**Review Assessment: Checking Correctness Of Experiments:**

I assessed the sensibility of the experiments.

**Review Assessment: Thoroughness In Paper Reading:**

I read the paper at least twice and used my best judgement in assessing the paper.

---

> ### Author Response · Authors · 2019-11-14
> **Response to your review**
>
> "The paper discusses a new strategy for deep semi-supervised learning that seems related to the deep label propagation method of Iscen et al. 2019, but is more scalable and has a different loss function."
>
> Thanks for summarizing our work. What you are saying is correct, though we would like to emphasize an important difference between LLP and Iscen et al. 2019.  LLP is not at all merely a more scalable implementation of the traditional label propagation idea used by Iscen et al.  It's actually a quite distinct algorithm for propagation, and it is these conceptual differences that are critical to its increased scalability. The key difference between LLP and Iscen et al. is that LLP leverages the "non-parametric softmax" embedding framework to propagate the labels from labeled examples to unlabeled examples. As a result, LLP formulates a novel label propagation algorithm via distances in the embedding space, using mathematically well-defined and efficiently computable (non-normalized) probability measures which are weighted by the local densities of the labeled examples in the embedding space. Because these local distributional computations are very simple but remain sensitive to subtle data statistics, LLP's newly proposed label propagation algorithm is at least as effective as, and much more scalable than, traditional label propagation schemes like that used in Iscen et al, enabling the application of LLP to large datasets that are inaccessible to those more traditional approaches.
>
> "With suboptimal preprocessing, it also performs comparable / slightly better than UDA (Xie et al. 2019) (The authors speculate it could do better with the preprocessing that UDA uses)"
>
> Thanks for mentioning this! We would like to emphasize a couple of points distinguishing LLP from UDA.
> The first point is that LLP operates on an extremely different idea from UDA. More specifically, LLP leverages the unlabeled examples through embedding them with labeled examples together in a shared embedding space that aggregates statistically similar unlabeled datapoints together with labeled (putative) counterparts. This enables LLP to exploit the subtle statistical relations between labeled and unlabeled examples. In contrast to this, UDA basically just uses the unlabeled examples to robustify the networks against data augmentations, and then uses a large-scale hyperparameter search over the data augmentation space to discover a strong augmentation scheme.  UDA also benefits from employing computationally expensive but practically high-impact details such as the use of an *extremely* large batch size during optimization. Thus it's not really clear how much the improvement in UDA is actually attributable to the discovery of better augmentations versus just being due to the more intensive optimization scheme (both probably play some role).
>
> Thus, in a sense, UDA --- unlike LLP --- doesn't really involve nontrivial semi-supervised learning at all. But the strong data augmentations and the optimization details that drive UDA are pretty orthogonal to what LLP does. LLP currently just uses standard preprocessing and standard training batch sizes, and so, as you note, could presumably leverage these techniques in a complementary way.  (The process of implementing this is somewhat extended engineering effort so we don't have results on this at the moment.)
>
> Finally, it is probably useful to add that with the "rate-jump" phase now in our training, LLP now surpasses UDA by significant margins. For ResNet-50 with $p=10$ and $q=100$, LLP achieves better top5 (89.55 v.s. 88.52, LLP v.s. UDA) and top1 (70.85 v.s. 68.66). Please refer to our general comment and the revised paper for more details.

---

### Author Response · Authors · 2019-11-14
**Highlights in our revision**

We want to first thank all reviewers for their comments. We've separately posted detailed responses to each reviewer's comments.

We've also updated our paper a bit, to reflect a minor but useful improvement in our training procedure. The improvements are due to a new learning rate schedule in which, after the initial rate drop schedule, learning rate is increased and the dropped again.  Actually, the suspicion that such a "rate jump" procedure might be beneficial was already indicated by the performance difference between models with $q=100$ and $q=100 + $FT in Table 3 of our original submission, so we've taken this to its logical conclusion.  (Because rate-jumping is a conceptually small difference in training procedure rather than a change in our main algorithm, we think it's a reasonably fair addition to the paper.)

After applying rate-jump, top1 accuracy (%) of ResNet-18 with 10% of ImageNet labels improves from 61.51 to 63.20 (with similar differences for 3% and 5% labels), further increasing the gap between LLP and previous state-of-the-art methods. Especially of note is the large performance increase for the smallest amount of labelled data (1% labelled), with top1 performance increasing from 27.14% to 33.55% for Resnet-18 and 61.89% to 72.20% for Resnet-50 top5 (now surpassing previous SOTA by almost 19%).  It is also perhaps worth mentioning that the gap between rate-jumped LLP and the very recent UDA method is also now substantial.

It is also interesting to note that, so far, only LLP seems to be able to take advantage of rate-jumping: as we note in the revised paper, trying rate-jumping either on standard supervised learning or other semi-supervised procedures such as Mean Teacher do not lead to performance gains.  Essentially, we think what's happening is that as LLP creates better embeddings, confidences and quality of pseudolabels increases substantially, to the point where there's enough signal that high learning rates again become useful. Especially, our first several drops of the learning rate usually lead to jumps in the performance and therefore significant improvement to the embedding quality. Nevertheless, this improvement requires a larger learning rate to be leveraged, which is why a later learning rate jump can help. This type of dynamic basically can only apply to distributional methods like LLP (rather than "point-wise" methods like MT, or UDA, which don't create pseudolabels). Thus, we think the new improved results a further indication of the power of the LLP method, and not just entirely an additional "trick". However, to ensure people can make fair comparisons to other works, we've updated paper to show results both for the standard training procedure and the rate-jump procedure, with the latter clearly annotated.

---

### Decision · Program_Chairs · 2019-12-19

**Decision:**

Reject

**Comment:**

The paper introduces an approach for semi-supervised learning based on local label propagation. While reviewers appreciate learning a consistent embedding space for prediction and label propagation, a few pointed out that this paper does not make it clear how different it is from preview work (Wu et al, Iscen et al., Zhuang et al.), in addition to complexity calculation, or pseudo-label accuracy. These are important points that weren’t included to the degree that reviewers/readers can understand, and reviewers seem to not change their minds after authors wrote back. This suggests the paper can use additional cycles of polishing/editing to make these points clear. We highly recommend authors to carefully reflect on reviewers both pros and cons of the paper to improve the paper for your future submission.